# Navigating the Spectrum: Assessing the Concordance of ML-Based AI Findings with Radiology in Chest X-Rays in Clinical Settings

**DOI:** 10.3390/healthcare12222225

**Published:** 2024-11-07

**Authors:** Marie-Luise Kromrey, Laura Steiner, Felix Schön, Julie Gamain, Christian Roller, Carolin Malsch

**Affiliations:** 1Institute for Diagnostic Radiology and Neuroradiology, University Medicine Greifswald, 17475 Greifswald, Germanyjulie.gamain@med.uni-greifswald.de (J.G.); christian.roller@med.uni-greifswald.de (C.R.); 2Institute and Polyclinic for Diagnostic and Interventional Radiology, Faculty of Medicine, University Hospital Carl Gustav Carus Dresden, TU Dresden, 01307 Dresden, Germany; felix.schoen@ukdd.de; 3Institute for Mathematics and Computer Science, University Greifswald, 17489 Greifswald, Germany; carolin.malsch@uni-greifswald.de

**Keywords:** artificial intelligence, thoracic radiography, X-ray, fracture, pleural effusion, pneumonia

## Abstract

**Background:** The integration of artificial intelligence (AI) into radiology aims to improve diagnostic accuracy and efficiency, particularly in settings with limited access to expert radiologists and in times of personnel shortage. However, challenges such as insufficient validation in actual real-world settings or automation bias should be addressed before implementing AI software in clinical routine. **Methods:** This cross-sectional study in a maximum care hospital assesses the concordance between diagnoses made by a commercial AI-based software and conventional radiological methods augmented by AI for four major thoracic pathologies in chest X-ray: fracture, pleural effusion, pulmonary nodule and pneumonia. Chest radiographs of 1506 patients (median age 66 years, 56.5% men) consecutively obtained between January and August 2023 were re-evaluated by the AI software InferRead DR Chest^®^. **Results:** Overall, AI software detected thoracic pathologies more often than radiologists (18.5% vs. 11.1%). In detail, it detected fractures, pneumonia, and nodules more frequently than radiologists, while radiologists identified pleural effusions more often. Reliability was highest for pleural effusions (0.63, 95%-CI 0.58–0.69), indicating good agreement, and lowest for fractures (0.39, 95%-CI 0.32–0.45), indicating moderate agreement. **Conclusions:** The tested software shows a high detection rate, particularly for fractures, pneumonia, and nodules, but hereby produces a nonnegligible number of false positives. Thus, AI-based software shows promise in enhancing diagnostic accuracy; however, cautious interpretation and human oversight remain crucial.

## 1. Introduction

Over the past years, there has been increasing effort to integrate artificial intelligence (AI) in radiology workflow, the notion being that the use of such algorithms enhances diagnostic accuracy and efficiency [1]. In times of manpower shortage in a challenging labor market, there exist practical limitations regarding full-time availability of expert radiologists, especially in rural areas and concerning overnight coverage [2,3]. According to the Clinical radiology census report by The Royal Society of Radiologists, there was a shortfall in clinical radiology consultants of 30% in 2023, which is forecasted to increase to 40% by 2028 [4].

One important, but by far not the only, application of AI software is the emergency setting, where quick diagnosis is essential for patient well-being. Here, AI may help speed up the reporting process and provide direct feedback to clinicians regarding relevant findings [5]. Besides the enhancement of diagnostic accuracy through image analysis and interpretation, AI has also high potential for workflow optimization and predictive analysis [6]. Nevertheless, although AI tools have been evolving for some years now, their application is still mostly restricted to large academic centers or general hospitals, and widespread clinical implementation is still lacking [7,8]. One, among other reasons, might be that the introduction of AI tools into clinical practice may not be welcomed by all radiologists—despite or precisely because of the obvious advantages mentioned before. A survey by Huisman et al. found that radiology residents and radiologists feared replacement by AI software, especially those doctors who had limited AI-specific knowledge levels [9]. However, the purpose of such AI tools is not to replace the radiologist but to integrate computer-generated information in the decision-making process [10]. In an ideal setting, radiologists independently examine the image and take the algorithm’s suggestions into account, potentially modifying their initial diagnosis where it seems necessary [11]. Synergistic effects of this approach have been shown previously [12,13]. Yet, there exists also the risk of automation bias—that is, the overreliance on a decision support system without critical revision [11,14], which may consequently lead to following up on a decision even if it was incorrect.

Conventional imaging techniques like chest radiographs are an ideal target for AI tools. A chest X-ray is the first-line examination for the evaluation of thoracic diseases and the most frequently requested radiology investigation [15,16]. However, it does not only generate a high number of datasets for software training and usage, but also causes a significant burden for healthcare systems. There exist several studies dealing with the performance of chest radiograph AI software, whereby the vast majority focused on specific diagnoses, such as tuberculosis or COVID-19 detection [17,18,19,20,21,22], malignancies [23,24,25] or pneumonia [26].Those studies mostly used enriched datasets, which differ from consecutive clinical data in terms of disease prevalence and population diversity. Their applicability, therefore, is limited. Only a few publications dealt with a wider range of diagnoses [27,28,29]. Furthermore, there remains the problem that among the over 500 FDA (Food and Drug Administration)-approved radiological AI applications the vast majority has not been tested in a clinical setting led alone by third parties [30]. According to a study by van Leeuwen et al., there was peer-reviewed evidence of efficacy in just 64 out of 100 CE-marked AI software products [31]. Only 18 AI products have demonstrated potential clinical impact, and half of the available evidence was (co-)funded or (co-)authored by the vendor. Consequently, there remains the demand for independent and peer-reviewed evaluation of existing AI application, especially in a clinical setting.

A standard approach for the evaluation of AI performance is using the radiologists’ evaluation as ground truth. However, even radiologists are not omniscient and often have different opinions concerning image interpretation. Therefore, we employed radiologists’ diagnoses in awareness of AI output as the reference standard in those cases where the primary rater and software disagreed.

The presented study investigates the performance of a deep learning-based AI software for interpreting chest radiographs in a consecutive clinical setting. The study has two primary objectives: First, to evaluate the concordance between diagnoses generated solely by AI-based software and those established within routine radiological practice, which is augmented by the additional use of AI. Second, to identify factors contributing to diagnostic inconsistencies. This two-step approach aims to clarify the potential value of AI-based software in the decision-making process of routine care while simultaneously quantifying the risk of overreliance on AI in clinical practice.

## 2. Methods

### 2.1. Study Design

A cross-sectional study was conducted at the Institute of Diagnostic Radiology and Neuroradiology of the University Medicine Greifswald, Germany. The study was approved by the local institutional review board (BB 139/19a), which also waived the requirement to obtain patients’ informed consent due to the retrospective study design. According to common ethical standards, patient confidentiality and data protection were ensured by pseudonymization with only the study coordinator having access to the subject identification list.

This study was carried out independently of the software provider Infervision Medical Technology Co., Ltd. (Beijing, China). The company was not involved in any aspect of the study design, data interpretation, or manuscript development.

The study was conducted in accordance with the principles of the Declaration of Helsinki and the STROBE (Strengthening the Reporting of Observational Studies in Epidemiology) guidelines.

#### 2.1.1. Study Population

The electronic database was queried for chest radiographs acquired between January and August 2023 meeting the following inclusion criteria: patients aged 18 years or older, chest X-rays taken in both posterior-anterior (PA) and lateral views while standing, with adequate image quality, and accompanied by a comprehensive written report detailing the diagnoses under investigation. In total, 1506 patients were included, covering those admitted to the emergency department as well as in-house patients.

To calculate the number of individuals required for the study, a power simulation was conducted based on 512 already released chest X-ray images from a previous study [32]. To estimate reliability measures such as Cohen’s kappa and confidence intervals with a reasonable performance, a sample size of 1500 was found to be sufficient.

#### 2.1.2. Radiograph Acquisition

All chest radiographs were acquired using Digital Diagnost C90 radiography units (Philipps Healthcare, Eindhoven, The Netherlands) in anterior-posterior and lateral views (tube voltage: 125 kV, tube-detector distance: 1800 mm, detector resolution: 2900 lines, 2456 columns). Images were acquired in DICOM format.

#### 2.1.3. Deep Learning Algorithm

All patient examinations were automatically transferred via the hospital’s intranet in DICOM format from the radiology department’s PACS to an independent on-site server. Subsequently, the AI diagnostic was conducted using the CE-certified commercial software InferRead DR Chest^®^ (Version 1.0, Infervision Medical Technology Co., Ltd., Beijing, China). Results were sent back in HTML file format, which can also be accessed via the intranet at the radiological workstations. The algorithm is based on a convolutional neural network (CNN), particularly the U-NET architecture. This is a subtype of CNN specifically designed for biomedical image segmentation [33]. The algorithm consisted of five contracting and expansive layers. The software was trained for the detection of 17 different thoracic abnormalities, the training set for each type of lesion consisting of over 20,000 annotated images. A detailed description of the algorism’s architecture has been published previously [32].

### 2.2. Image and Data Analysis

Four selected major diagnoses were chosen for investigation: fracture, pleural effusion, lung nodule, and pneumonia. All fractures that are visible in the chest X-ray were included (this includes the clavicle, ribs, shoulder, and upper arm). Pleural effusions visible in the lateral or PA view were considered. Lung nodules, by definition, included all findings under 3 cm in diameter; larger findings were classified as lung masses and not included in the study. The diagnosis of pneumonia in this study encompasses all three radiological manifestations (bronchopneumonia, lobar pneumonia, interstitial pneumonia).

#### Image Analysis

The primary gold standard for radiological diagnostics at the study site is a written report, in which all chest radiograph images are initially evaluated during clinical routine by radiology residents with subsequent approval by a senior radiologist according to the four-eyes principle. For image interpretation the software DeepUnity Diagnost 1.2.0.1^®^ (Dedalus HealthCare, Bonn, Germany) was used. In order to rule out biases caused by differences in image quality which can, for instance, be related to the type of image equipment used, only the same DIN-certified diagnostics monitors (Barco MRXT 4700, Barco N.V., Kortrijk, Belgium) were used for image evaluation. Those monitors routinely undergo a daily check to ensure constant imaging quality by the radiologist him- or herself, and a more advanced control every six months by a specialized medicine physician. The initial evaluators did not have access to the AI tool. For the present study, all diagnoses of interest were extracted from written reports as binary variables. Information on age and gender of the patients was documented.

Mirroring the standard human diagnostic process, all written reports meeting the eligibility criteria were processed with the AI software. An example output is shown in Figure 1. These diagnoses were extracted by a medical student as binary variables and represent the first rating to be analyzed in this study.

Any inconsistencies between the radiologist’s written report and the AI decision were marked for re-evaluation by two radiologists specialized in thoracic imaging in awareness of the AI’s results. They reached a consensus decision in case of discrepancies. Revised diagnoses of the medical reports (i.e., radiologist’s diagnoses augmented by AI findings) were employed as the second rating in this study.

### 2.3. Statistical Methods

Data were recorded in Microsoft Excel 2021^®^ (Microsoft Inc., Redmond, WA, USA). All statistical analyses and computations were performed in R version 4.3.2. Absolute and relative frequencies of binary or categorial variables were reported, and groups were compared using a chi-squared or Fisher’s exact test. *p*-values below 0.05 were considered statistically significant. Adjustment for multiple testing is not carried out due to the exploratory character of the study. The agreement of binary diagnoses (positive/negative) was assessed using absolute agreement percentage, which reflects the consensus between both raters regarding the diagnosis. To adjust for agreement by chance, Cohen’s kappa coefficient was calculated together with bootstrapping confidence intervals for a confidence level of 95%. Since Cohen’s kappa is sensitive to imbalances in the contingency table and might result in under-estimation, prevalence and bias indices, as well as PABAK estimates [34] were calculated (prevalence-adjusted bias-adjusted kappa, as described in [35] with bootstrapping confidence intervals). Furthermore, Gwet’s AC1 coefficient was applied to account for possible bias effects between raters, i.e., when raters might have different probabilities of assigning categories [36,37].

The reliability analysis was stratified for gender.

## 3. Results

### 3.1. Patient’s Characteristics

In total, 2267 individuals were initially identified in the electronic database. Out of those, 761 had to be excluded due to reasons such as radiographs being taken in a supine position, incomplete findings, or poor image quality. Finally, the study included 1506 patients, comprising both emergency department admissions and in-house patients. Among these, 851 (56.5%) were male and 655 (43.5%) female. The median age was 66 years with an interquartile range of 56–76. The majority of patients were aged 60 years or older (68.9%), with age group distributions as follows: 0–19 years: 10 (0.7%), 20–39 years: 154 (10.2%), 40–59 years: 305 (20.3%), 60–79 years: 750 (49.8%), and 80 years or older: 287 (19.1%).

### 3.2. Diagnoses

In 860 (57.1%) patients, no inconsistencies in the diagnoses by AI alone and radiologists were found. However, in the remaining 42.9% of cases, inconsistencies were identified. This includes 500 cases (33.2%) with inconsistencies in one of the four diagnoses, 135 cases (9.0%) with inconsistencies in two, and 11 cases (0.7%) with inconsistencies in three out of the four diagnoses. No difference in the number of inconsistent diagnoses was found between genders (*p* = 0.183 in Fisher’s exact test).

The AI software reported notably more findings compared to radiologists augmented by AI (1116 (18.5%) vs. 671 (11.1%) of all ratings (see Table 1). It diagnosed fractures, pneumonia and nodules significantly more often than radiologists augmented by AI (18.1% vs. 6.5%, 28.0% vs. 11.9% and 16.5% vs. 5.6%, respectively, all *p* < 0.001). Conversely, radiologists augmented by AI diagnosed pleural effusions significantly more often than AI software alone (20.6% vs. 11.5%, *p* < 0.001). Overall, the number of positive findings identified by AI alone is 1.7 times higher than those identified by radiologists using AI assistance. Specifically, AI detects 2.8 times more fractures, 2.4 times more cases of pneumonia, 3.0 times more nodules, but only 0.6 times as many pleural effusions. Absolute agreement of ratings (including positive and negative findings) through radiologists and AI software was highest for pleural effusions (1357, 90.1%) and lowest for pneumonia (1237, 82.1%).

A detailed breakdown of the frequencies of diagnoses made by radiologists augmented by AI versus AI software alone can be found in Figure 2 and Figure 3. The AI software identified 173 (11.5%), 189 (12.5%), and 256 (17.0%) cases for nodules, fractures, and pneumonia, respectively, as positive, whereas the radiologist assessed them as negative, being aware of the AI report. Conversely, the radiologist made 8 (0.5%), 15 (1.0%), and 13 (0.9%) positive diagnoses, respectively, that were rated as negative by the AI. Diagnostic agreement was achieved in 11% of diagnoses for pneumonia and 11.1% of diagnoses for pleural effusion; however, in only 5% and 5.5% for nodules and fractures, respectively.

### 3.3. Reliability Analysis

Cohen’s kappa, PABAK and Gwet’s AC1 reliability estimates are presented in Table 2 and Figure 4. Reliability estimates are smallest for Cohen’s kappa [38] and increase when using PABAK or Gwet’s AC1, indicating that the moderate agreement shown by Cohen’s kappa may be distorted by a high prevalence for certain diagnoses or exhibit a slight bias in the ratings. Reliability is highest for pleural effusions. The lowest reliability varies by method, with fractures (0.39, 95%-CI 0.32–0.45) and nodules (0.41, 95%-CI 0.34–0.48) showing the lowest agreement using Cohen’s kappa, while pneumonia has the lowest reliability using PABAK and Gwet’s AC1.

No consistent gender differences were observed. Confidence intervals of Cohen’s kappa are wider for females than for males due to the smaller number of observations in the sample.

The increased prevalence indices shown in Table 2 explain why PABAK estimates yield notably higher reliability than Cohen’s kappa. However, PABAK eliminates all prevalence and bias effects, which hold informative value in our analysis. The balanced prevalence would imply equal numbers of concordant ratings for both the presence and absence of a pathology. Assuming all ratings are in concordance (bias index = 0), this would imply that the prevalence of each pathology is 0.5, which contradicts the nature of most pathologies.

### 3.4. Inconsistency Analysis

The causes of AI software algorithms generating false diagnoses are varied.

First, and most important, there is the problem of so-called shortcut learning, which is that AI algorithms rely on spurious correlations rather than on true pathological features [39]. An AI algorithm, for instance, might learn to associate certain external material (like oxygen hoses and chest tubes) or also demographic features with a specific diagnosis instead of recognizing the actual pathological features of a disease.

Another cause of malperformance by AI software arises from data bias. AI algorithms heavily rely on the quality, quantity and diversity of their training data. However, the images used for training often represent specific demographic or geographic populations, so that historically underserved groups, such as females or racial minorities, are subject to misinterpretation [40]. Similarly, training with narrow datasets that do not encompass a variety of diseases lead to problems in effectively generalizing to new cases encountered in clinical settings [41].

Examples of discrepancies between AI and radiologists are given below.

#### 3.4.1. Fractures

A frequent mistake made by the AI is interpreting the joint surfaces of the shoulder depicted in different projections, as well as projections of the scapula onto the thorax, as fracture lines due to their alignment with the thorax. False positive findings can also occur because of the superimposition of bones with foreign material, such as catheters.

On the other hand, vertebral fractures are most often overlooked by the AI because these are best assessed in the lateral view, whereas the software only takes the PA view into account. Examples of false positive and negative osseous findings are shown in Figure 5.

All in all, thoracic fractures are a subset of pathologies which are often overlooked by radiologists (see Figure 5D), who rather turn their attention to lung pathologies—a fact which is substantiated by our results showing a high number of fractures detected by AI alongside a low agreement with the human rater (see Figure 3).

#### 3.4.2. Pneumonia

False-positive AI findings occur in younger patients with dense breast tissue, as well as with increased vascular markings indicative of congestion (Figure 6A). The AI also frequently interprets poorly defined lung masses as pneumonic infiltrates.

False-negative results for pneumonia often occur in cases of early pneumonic infiltration or when the infiltrates are located retrocardially. Similarly, pneumonia near the hilum is frequently undetected by the software (Figure 6B).

#### 3.4.3. Nodules

Besides fractures, the detection of lung nodules also shows high discrepancies between software and human raters. Smaller nodules are sometimes overlooked by radiologists (Figure 7B), whereas pseudo-lesions—structures such as nipple shadows, catheter material or port needles as well as external foreign bodies—are sometimes falsely interpreted as pulmonary nodules by the AI (Figure 7D). These pitfalls occur only seldom with radiologists, who can easily correctly identify these items for what they really are. Similarly, transversely sectioned vessels or bronchi are diagnosed as round opacities indicative of pulmonary nodules due to their morphology. For pleura-adjacent nodules, the distinction from pleural plaques is not always clear.

False-negative findings occur, for example, due to the misinterpretation of a pulmonary nodule as pneumonia because of overlapping morphology. Additionally, the diagnosis of masses in the lingula segment of the lung is complicated by the overlap with the heart.

#### 3.4.4. Pleural Effusions

Differences in the diagnosis of pleural effusion can be explained as follows: Small amounts of a pleural effusion are best identified as a dorsal costophrenic angle effusion in the lateral projection. However, AI primarily uses the PA projection for diagnosis, which means that smaller amounts of effusion might be overlooked (Figure 8).

## 4. Discussion

The study assessed the performance of the AI-based diagnosis support software InferRead DR Chest^®^ in a clinical setting. We compared AI-generated diagnoses alone with those from radiologists’ expertise augmented by AI and found that AI software alone generates a relevant number of false positives. This specifically holds for fractures, pneumonia and pulmonary nodules, where the number of positive findings by AI software alone was 2.4- to 3-fold compared with radiologists augmented by AI. For pleural effusions, however, radiologists identified a substantially higher number of cases, indicating a noteworthy underperformance of AI. Reliability of ratings was highest for pleural effusions, demonstrating at least moderate to good agreement. For the remaining pathologies, reliability was found to be no more than moderate. These findings emphasize the potential risk of excessive reliance on decision support systems without thorough critical review.

### 4.1. Preliminary Methodological Thoughts

Before discussing the results in a clinical context, it is important to address several methodological aspects of our study design and analysis. We used various reliability metrics such as the Phi coefficient, Fleiss’ kappa, and Krippendorff’s alpha for validation of the Cohen’s kappa estimates, all of which yielded consistent results. Cohen’s kappa and other metrics, however, are sensitive to imbalances in the contingency table. Therefore, we expect negative bias in these estimates. To account for this, we additionally calculated PABAK and Gwet’s AC1 estimates, resulting in notably higher values of agreement. However, eliminating all prevalence and bias effects seems to contradict the nature of most pathologies and seems to introduce positive bias. A simulation study characterizing possible distortion mechanisms is beyond the scope of this study. Therefore, and given the fact that for every third patient a discrepancy was observed in at least one of the diagnoses, we will rely on the lower reliability estimates produced by Cohen’s kappa for interpretation and consider them a conservative estimation of the ground truth. Moreover, there is no reliability estimator that perfectly matches the needs for our analysis, since reliability estimation assumes independence of ratings. This is not given in our study, since the radiologist’s ratings include knowledge of the AI. Hence, all estimations resulting from studies with non-independent ratings need to be interpreted cautiously.

### 4.2. Clinical Implications

Since chest radiography remains one of the most common imaging examinations performed worldwide, the use of AI tools offers promising perspectives to speed up and enhance the quality of radiology reports. No algorithm, however, can be completely foolproof, so there always remain false diagnoses made by AI. Most tools for chest X-ray analysis have the immanent problem that they only take the anterior-posterior image into account for decision making. The radiologists, also considering the lateral view, has thus an additional source of information in ambiguous cases. In our study, we found the overall number of positive findings identified by AI alone to be 1.7 times higher than those identified by radiologists using AI assistance. False-positive findings, however, do have an enormous impact on patient well-being as well as the healthcare system itself: For the patient, a false-positive diagnosis may lead to unnecessary and potentially harmful follow-up diagnostic or even therapeutic procedures. For instance, a pseudolesion mistaken as a pulmonary nodule by the AI may result in an additional computertomographic clarification, which comes along with an unnecessarily high radiation dose. A wrongly identified pneumothorax may in the worst case result in the placement of a chest tube with all its risks and discomfort for the patient. Apart from that, imaging findings are often associated with psychological stress for the patient [42]—in the case of a false-positive diagnosis, this is even more gratuitous. On top, every additional diagnostic or therapeutic procedure puts a financial burden on already strained healthcare systems.

AI is often proclaimed to be the solution to manpower shortages, aiding the inexperienced radiology resident or clinician in image interpretation at hospitals where expert radiologists are absent. Yet, against the background of the above-mentioned implications, this has to be seen critically. An overreliance on decision support systems without critical revision by an expert must be averted by all means. This so-called automation bias has long been the subject of research, not only in medicine. A number of studies, for instance, have shown that the introduction of computer-aided detection into mammography workflow impairs the performance of radiologists [11,43,44]. All radiologists, regardless of their expertise, can be subject to automation bias; however, inexperienced readers are more prone to follow an incorrect suggestion made by AI. Introducing AI support systems at an early stage of radiology training, therefore, may bare the risk of deskilling in a way that essential skills are lost or not properly learned [11]. Strategies to handle automation bias effects may include giving confidence levels to the decision support system [45] and educating readers about the reasoning process of the software [46].

Radiology, as a highly technology-affine medical branch, has always been a vanguard for technical improvement and development. Roughly three-quarters of the over 700 medical AI applications approved by the FDA are implemented in radiology departments [30]. However, the vast majority have not been tested in a clinical setting, let alone by third parties. Kim et al. found that, from 516 published studies, only a minimum of 6% underwent external validation [47]. An external validation in a population different from the training population, however, constitutes the important feature to verify an algorithm’s ability to generalize across the expected variability of real-world data in different hospital systems. Deep learning algorithms depend on a large quantity of high-quality labelled data for training. The most relatable explanation for the lack of proper validation is that producing this magnitude of medical image data is resource-intensive, so that developers rely on convenience case-control data. An algorithm trained only at a single institution will, however, most probably provide inaccurate outputs when run at other hospitals [48,49].

According to an international survey, other hurdles to the implantation of AI software were most often organizational, like ethical and legal issues, workflow integration or cost-effectiveness, as well as personal factors, such as lack of knowledge [50]. Limited AI-specific knowledge leads to increasing fear and decreasing levels of an open and proactive attitude towards the topic. Here, at least each institute can take appropriate action: Teaching on correct handling AI software should be part of internal training rounds and be integrated into residency programs with caution. Each radiologist should retain critical oversight when incorporating AI tools in their workflow and be aware of automation biases.

### 4.3. The Study in the Context of Existing Literature

Apart from these hurdles, another shortcoming of existing AI tools is that most applications are designed to perform specific tasks only and are, therefore, limited in detecting a wide range of pathologies [51]. Some previous studies avoid this shortcoming by discriminating only between normal and abnormal conditions, skipping any further differentiation of pathologies [52]. While this approach bears some potential in terms of a triage, channeling the radiologist’s attention towards pathological radiographs, it leaves a substantial amount of workload for the radiologists. A comprehensive aid in interpreting pathological scans is of much more value in the clinical routine.

A meta-review of 59 papers on deep learning algorithms for automated detection of pneumonia in chest X-ray images indicated high diagnostic performance (pooled sensitivity, specificity and accuracy of 0.98, 0.94 and 0.99, respectively) [53]. Similar results can be observed concerning pulmonary nodules and pleural effusion [5,54], whereas studies covering fractures on chest X-ray are negligible. These very good outcome values in other studies often derive from the use of curated datasets including a higher percentage of pathologies than seen in general clinical practice, which leads to an artificially high accuracy. A strength of our study is the usage of consecutive data from a real-world clinical setting, containing emergency as well as in-house patients.

The same software used in our study was investigated by Peters et al. [55] on chest phantom radiographs. The authors reported an interrater reliability of 0.51 (Fleiss’ kappa) for the detection of lung nodules between radiologists in training with varying levels of expertise and AI. These results show moderate agreement between radiologists and AI but differ from our results of 0.41 (CI 0.34–0.48). The authors report that they adjusted the algorithm to the site-specific characteristics of the radiography unit prior to deployment, which may have enhanced its reliability. Schweikhard et al. [32] extensively discussed this issue in their study involving 500 patients, which preceded our own. However, the validity of aligning such cutoffs with a specific regional cohort remains questionable. Although these cutoffs may be estimated from a sample of 500 images, as in the study by Schweikhard et al., their application to a larger dataset may not be justified. There is no guarantee that these cutoffs will remain valid over time, particularly in light of potential system or software changes. The opaque nature of these algorithms complicates their interpretation, especially when alterations to the framework occur.

Further studies have explored the reliability of different raters in interpreting radiographic images, but their designs are highly heterogeneous. This heterogeneity includes (a) the selection of raters: one or multiple raters, with different experience levels, working individually or together are compared with AI; (b) the comparison groups: clinicians alone versus AI alone or versus radiologists combined with AI; (c) the metrics used to assess agreement vary considerably; (d) the data sources: curated datasets specifically chosen for analysis, phantom models with pathologies generated for the study, retrospective data; (e) the algorithms employed, which range from custom-developed models to commercial software, some of which are regionally adjusted; and (f) the sample sizes. These factors complicate the comparison of individual study outcomes. Nevertheless, we will highlight two studies whose designs are reasonably comparable to our own.

In a prospective multicenter study on the performance of chest X-rays using a commercial AI, Vasilev et al. [56] found a kappa value between AI and radiologists of 0.42 (CI 0.38–0.45). Though images were only classified as either normal or pathological, results are comparable to our kappa values, which range between 0.39 and 0.63.

Pham et al. [57] developed an AI model (VinDr-CXR) for evaluating chest X-rays and assessed its practical applicability in a retrospective study by comparing it against three radiologists. Their Cohen’s kappa values of 0.49 (CI 0.32–0.66) between that for the best reader and AI for detecting nodules, indicating moderate agreement, is comparable to our results (0.41; CI 0.34–0.48). However, their kappa values for detecting pneumonia and pleural effusion were significantly higher: 0.74 (CI 0.62–0.85) versus our 0.46 (CI 0.41–0.51) for pneumonia, and 0.92 (CI 0.85–0.99) versus our 0.63 (CI 0.58–0.69) for pleural effusion. Despite these strong results, the wide confidence intervals suggest a degree of imprecision. Additionally, it is noteworthy that while each radiologist’s agreement with the AI was assessed, the inter-rater agreement among the three radiologists was relatively low, with an average Fleiss’ kappa of 0.40 (CI 0.33–0.48).

### 4.4. Limitations

Several limitations need to be acknowledged in our study. First, the study was conducted at a single institution, potentially limiting the generalizability of our findings to other healthcare settings. However, the study itself should be seen as a (vendor-independent) external validation of an already CE-certified AI software. Previous studies at other institutes have likewise reported on InferRead DR Chest^®^ from Infervision, which is why we refrained from a multicenter design for our study. Second, our study focused on a specific set of thoracic diagnoses, potentially limiting its representation of the broader spectrum of conditions encountered in clinical practice. However, generalizability was not our primary goal. Instead, this study serves as a counterexample, aiming to caution against overreliance on AI diagnoses without critical review of the results. Third, reliability estimation is not straightforward due to the chosen study design.

Lastly, the AI software’s algorithm classifies diagnoses based on a probability cutoff, which could introduce classification errors depending on the choice of the cutoff value. However, the probabilities generated by the deep learning algorithm and the cutoff values are not accessible to users in the current version of the software. A detailed discussion of data from the same institution is provided by Schweikhard et al. [32].

### 4.5. Conclusions

The findings of our study suggest that AI-based software has the potential to augment radiological diagnosis. However, the observed inconsistencies between AI and radiologists, especially the high number of false positives generated by AI, underscore the need for cautious interpretation of AI outputs and the importance of human oversight in the diagnostic process. Further research, for instance on the impact of automation bias in chest X-ray interpretation, is warranted to explore the clinical impact of integrating AI-based software into radiology practice. Against the background of mostly insufficient (external) validation of commercially available AI applications, there remains the demand for independent and peer-reviewed evaluation, especially in clinical settings.

## Figures and Tables

**Figure 1 healthcare-12-02225-f001:**
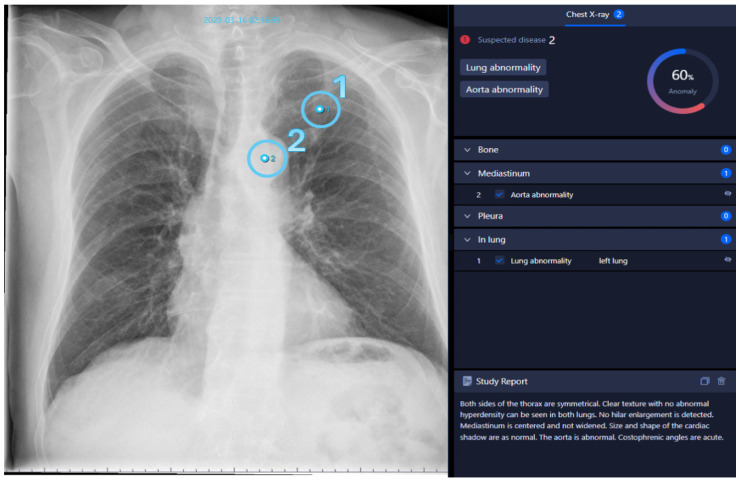
Chest radiograph analyzed with AI. AI-generated report based on a routine chest radiograph marking two findings together with their precise locations: one lung abnormality (1) and one aortic abnormality (2). All findings are listed on the right side, along with an overall abnormality probability score of 60%. Below the list is a concise report created by the software, similar to a radiology report.

**Figure 2 healthcare-12-02225-f002:**
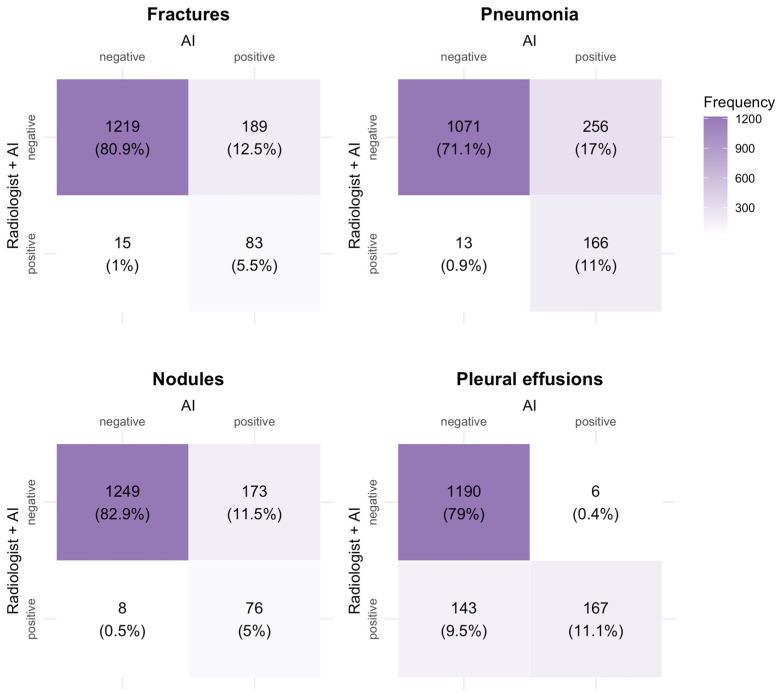
Frequency of diagnoses by radiologist augmented by AI versus AI only.

**Figure 3 healthcare-12-02225-f003:**
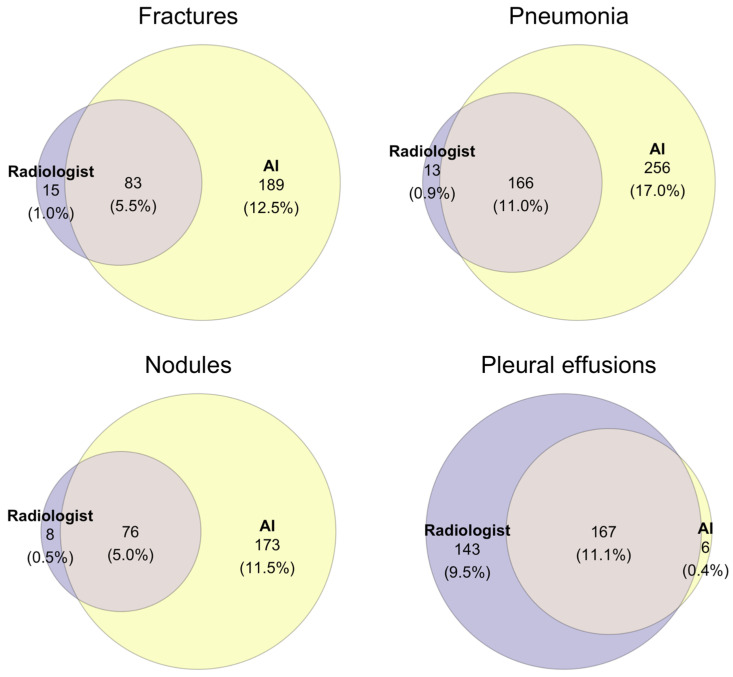
Frequency of positive diagnoses by radiologist augmented by AI (purple), AI software only (yellow) and agreeing diagnoses (intersection). Total sample size was 1506.

**Figure 4 healthcare-12-02225-f004:**
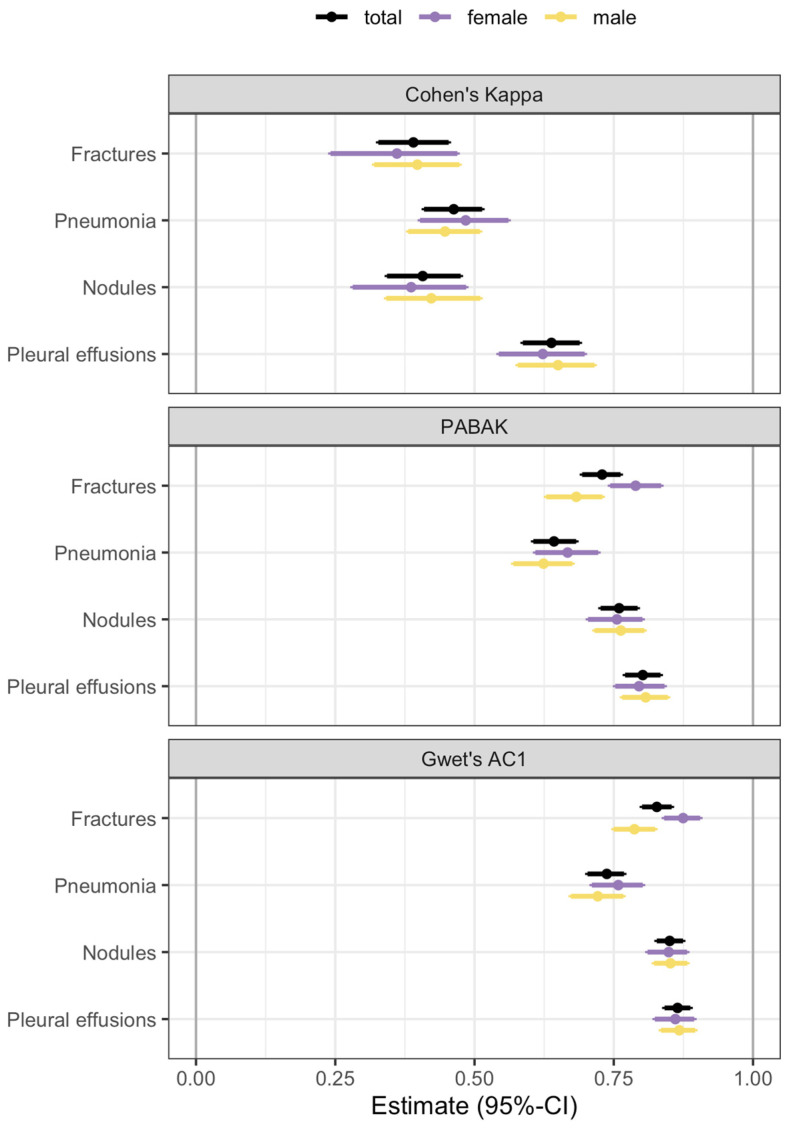
Estimates for Cohen’s kappa, PABAK and Gwet’s AC1 coefficient with 95% bootstrap confidence intervals for the total sample and stratified by gender.

**Figure 5 healthcare-12-02225-f005:**
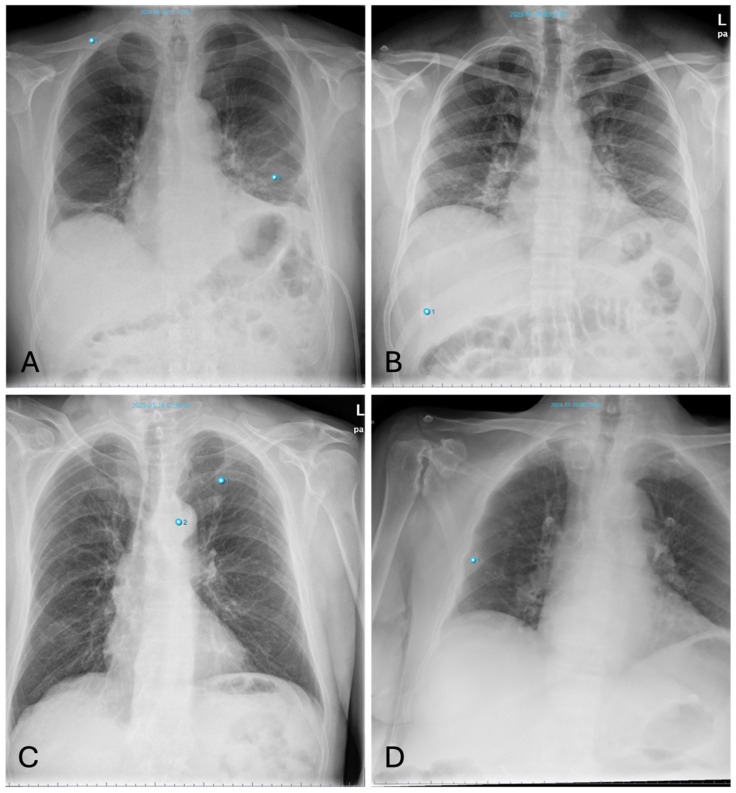
Inconsistencies of fracture diagnoses: (**A**)—false-positive fracture of the right clavicle described by AI at the overlay of the clavicle and 2nd rib (pneumonia in the left basal lung was correctly identified); (**B**)—false-positive fracture description of the 10th rib on the right by AI due to overlay of external oxygen hose; (**C**)—false-negative bone status; fracture of the left clavicle was not described by AI (Note: other finding incorrectly labelling “aortic abnormality”); (**D**)—rib fracture on the right found by AI but overlooked by radiologists.

**Figure 6 healthcare-12-02225-f006:**
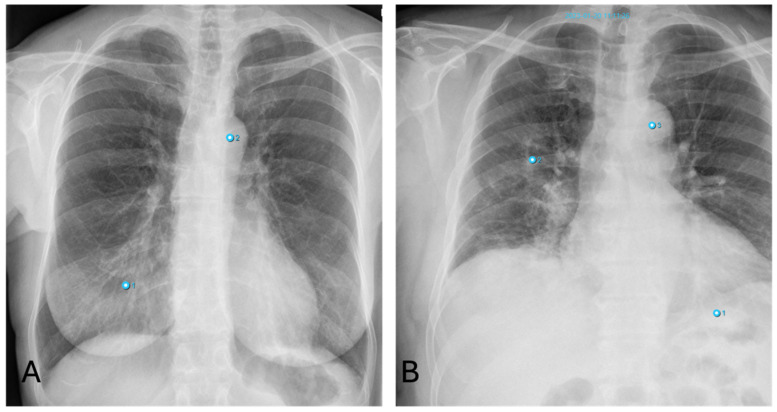
Inconsistencies of pneumonia diagnoses: (**A**)—false-positive AI interpretation of dense breast parenchyma as pneumonia in a young woman (Note: other finding incorrectly labelling “aortic abnormality”); (**B**)—false-negative interpretation of pneumonia. X-ray shows pneumonic congestion in the right lower lobe not detected by AI (Note: other findings incorrectly labelling “aortic abnormality” and “pneumonia”).

**Figure 7 healthcare-12-02225-f007:**
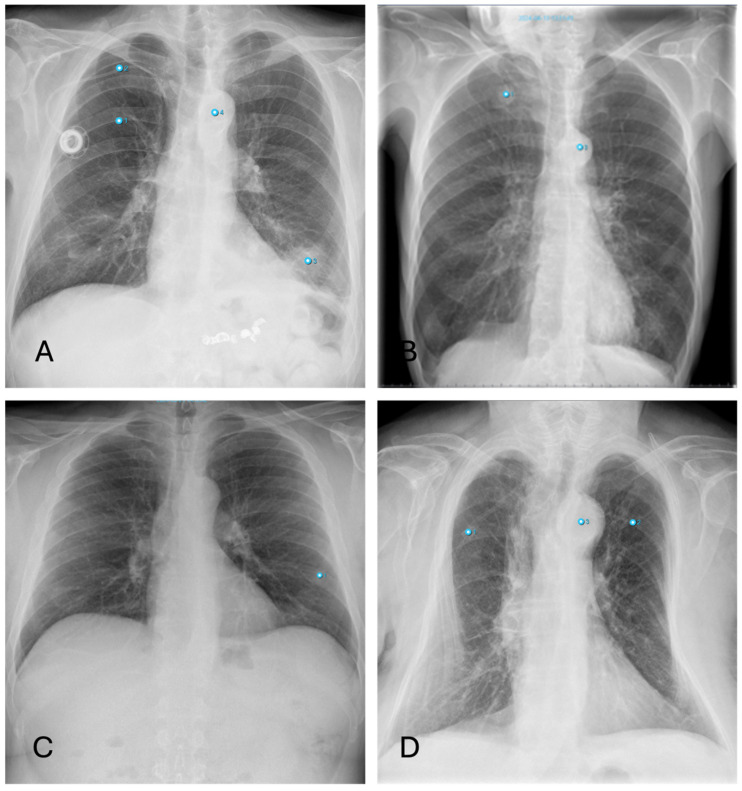
Inconsistencies of nodule diagnoses: (**A**)—false-positive AI interpretation of pulmonary mass in the lower lobe on the left as infiltration caused by pneumonia. (Note: Also false classification of Portcatheter as pneumothorax, again “aortic abnormality” and “rip fracture”); (**B**)—correctly identified nodule by AI overlooked by radiologist (Note: other finding incorrectly labelling “aortic abnormality”); (**C**)—False-positive nodule described by AI, really corresponding to the overlay of two ribs; (**D**)—False-positive nodule described by AI, really corresponding to external oxygen hose (Note: other findings incorrectly labelling “aortic abnormality” and “rib fracture”).

**Figure 8 healthcare-12-02225-f008:**
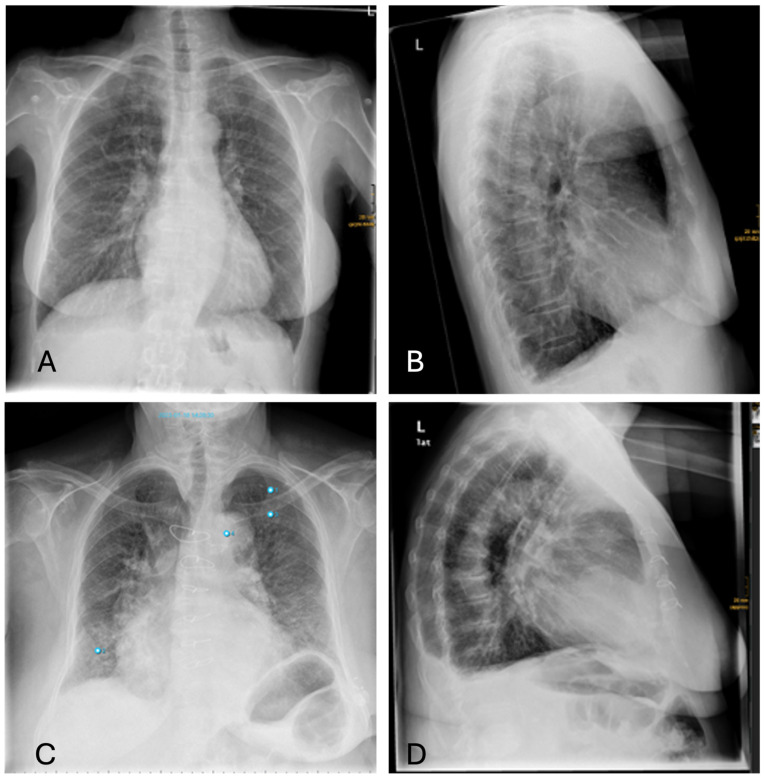
Inconsistencies of pleural effusion diagnoses: (**A**,**B**)—AI missing the sulcal effusion described by the radiologist based on the lateral projection; (**C**,**D**)—Similar case with an even more prominent pleural effusion not detected by AI (Note: other findings incorrectly labelling “aortic abnormality”, “pneumonia” and “rib fracture”).

**Table 1 healthcare-12-02225-t001:** Frequency of positive findings and agreement percentage through AI software alone versus radiologist augmented by AI.

	AI Software	Radiologist + AI	*p*-Value *	Agreement	Ratio **	PI ***	BI ****
Fractures	272 (18.1%)	98 (6.5%)	<0.001	1302 (86.5%)	2.8	0.75	0.12
Pneumonia	422 (28.0%)	179 (11.9%)	<0.001	1237 (82.1%)	2.4	0.60	0.16
Nodules	249 (16.5%)	84 (5.6%)	<0.001	1325 (88.0%)	3.0	0.78	0.11
Pleural effusions	173 (11.5%)	310 (20.6%)	<0.001	1357 (90.1%)	0.6	0.68	0.09
Overall	1116 (18.5%)	671 (11.1%)	<0.001	5221 (86.7%)	1.7	-	-

* Chi-squared test, ** Number of positive findings by AI software divided by number of positive findings by radiologists augmented by AI, *** PI = prevalence index, **** BI = bias index.

**Table 2 healthcare-12-02225-t002:** Estimates for Cohen’s kappa and 95% confidence intervals for the total sample and gender subgroups.

Diagnosis		Total	Males	Females
Fractures	Cohen’s kappa	0.39 (0.32–0.45)	0.40 (0.32–0.47)	0.36 (0.24–0.47)
	PABAK	0.73 (0.69–0.76)	0.68 (0.63–0.73)	0.79 (0.75–0.83)
	Gwet’s AC1	0.83 (0.80–0.85)	0.79 (0.75–0.82)	0.87 (0.84–0.90)
Pneumonia	Cohen’s kappa	0.46 (0.41–0.51)	0.45 (0.38–0.52)	0.48 (0.40–0.56)
	PABAK	0.64 (0.60–0.68)	0.62 (0.57–0.67)	0.67 (0.61–0.72)
	Gwet’s AC1	0.74 (0.70–0.87)	0.72 (0.67–0.76)	0.76 (0.71–0.81)
Nodules	Cohen’s kappa	0.41 (0.34–0.48)	0.42 (0.33–0.51)	0.39 (0.28–0.49)
	PABAK	0.76 (0.73–0.79)	0.76 (0.72–0.81)	0.76 (0.71–0.80)
	Gwet’s AC1	0.85 (0.83–0.87)	0.85 (0.82–0.88)	0.85 (0.81–0.88)
Pleural effusions	Cohen’s kappa	0.63 (0.58–0.69)	0.65 (0.58–0.71)	0.62 (0.55–0.70)
	PABAK	0.80 (0.77–0.83)	0.81 (0.77–0.84)	0.80 (0.75–0.84)
	Gwet’s AC1	0.86 (0.84–0.89)	0.87 (0.84–0.90)	0.86 (0.83–0.89)

## Data Availability

An R-package link for statistical analyses is provided in Section 2. Other research data are available from the authors by reasonable request.

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
