# Peer review of "Navigating the Spectrum: Assessing the Concordance of ML-Based AI Findings with Radiology in Chest X-Rays in Clinical Settings"

_healthcare, 2024, doi:10.3390/healthcare12222225_

Round 1

Reviewer 1 Report

Comments and Suggestions for Authors

The cross sectional study assesses the concordance between diagnoses made by a commercial AI-based software and conventional radiological methods augmented by AI for four major thoracic pathologies in chest X-ray: fracture, pleural effusion, pulmonary nodule and pneumonia.

The study reported that AI software detected thoracic pathologies more often than radiologists (18.5% vs. 11.1%). In detail, it detected fractures, pneumonia, and nodules more frequently than radiologists, while radiologists identified pleural effusions more often. Reliability was highest for pleural effusions (0.63, 95%-CI 0.58-0.69), indicating good agreement, and lowest for fractures (0.39, 95%-CI 0.32-0.45), indicating moderate agreement.

It concluded that AI-based software showed promise in enhancing diagnostic accuracy, for fractures, pneumonia, and nodules.

In general, the manuscript is well formatted and can contribute positively to the radiology and imaging techniques, only minor revisions may need to be considered.

Many typing mistakes should be revised (examples shown in Error! Reference source not found. Ex. Line 216).

Statistical analysis to be improved by revising Cohen’s Kappa reliability.

The study should consider critical false positive AI findings occurrence.

Conclusions needs more improvement.

References are OK and appropriate.

The tables and figures are well resented.

Comments on the Quality of English Language

it is OK but needs minor revision.

Reviewer 2 Report

Comments and Suggestions for Authors

In general, the manuscript is well written and well-structured. However, I have a few comments:

Abstract:

No corrections or comments.

Introduction:

Line 41-42: The purpose of the AI mentioned seems to lack a reference. Please include a suitable citation.

Materials and Methods:

Line 98: The address for Philips Healthcare is incorrect. The city should be listed as Eindhoven, not Amsterdam.

In the Methods section, the authors did not provide details about the reporting environment or the specifications of the diagnostic monitor used. It would be beneficial to include a brief description of the environment and the specifications of the monitor used to interpret the chest images.

 Line 138: The numbers for the different objects (areas of the images) in Figure 1 are too small and mostly unreadable.

 Results: Line 296: The figure reference should be Figure 7, not Figure 6, to avoid duplication of figure numbers.

The numbers on the chest images in Figures 4 to 7 are also too small, making them difficult to read clearly. Discussion:

No comments.

Conclusion:

No comments.

References: The text size and font style appear different from the rest of the manuscript. Please ensure consistency throughout..

Reviewer 3 Report

Comments and Suggestions for Authors

The current study investigates the concordance of commercial AI-based software and radiologists in diagnosing thoracic pathologies of chest X-rays of 1,506 patients. It observed that AI more often detected conditions like fractures, pneumonia, and nodules, while for pleural effusions, radiologists were better positioned. Overall, the finding underlines the potential of AI for improving diagnostic performance but also reminds us of the need for human oversight. However, some points should be clarified and described more to improve the quality of the article.

Recommendations:

Abstract:

1.      Background: If the background could be enriched with some existing challenges faced in radiology, such as a shortage of radiologists, that would be a positive contribution.

2.      Methods: Provide additional detail on the study design, including how the performance of the AI was compared to radiologists. Mention inclusion and exclusion criteria, and what were the statistical methods of analysis.

3.      Results: Give a brief description of the clinical importance of such findings.

Main text:

Introduction:

4.      Consider breaking up long sentences into shorter ones for better readability. It would also help to add some statistics relating to radiologist shortages or examples of how AI has already been successfully integrated into the clinic.

5.      Perhaps it would be good to relate how these objectives may have ramifications for clinical practice.

6.      More detailed citations or summarization of major studies would provide a stronger rationale or justification for the necessity of the study.

7.      Also, pointing out more explicit gaps in the literature would make the justification of this research stronger.

8. A very brief discussion of how the study will address the potential biases or limitations regarding reliance on radiologists diagnoses may be added.

Method:

9.      Although it states the approval of the institutional review board, the study does not elaborate on the ethical considerations that may have been considered to be sufficient for the waiver of consent. For instance, there should have been an elaborative discussion on how measures of patient confidentiality and data protection are followed to ensure strict adherence to ethical considerations.

10.  Lack of detailed demographic data about the study population limits the generalization of the results because of the. The age, gender, and comorbidity profile would likely give a better understanding of the sample's representativeness and applicability to broader clinical contexts.

11.  The method section does not discuss the possible biases related to the type of imaging equipment used. Differences in image quality and diagnostic capabilities can also occur when comparing different machines, which could affect study findings; there needs to be a discussion on how this is controlled or mitigated.

12.  The architecture of the algorithm is described, but no information on how this was validated against a benchmark dataset is given. This alludes that, by not doing so, there may indeed be a problem with the reliability and performance of this algorithm in a clinical setting. A suitably developed discussion of all the processes involved in validation and metrics would enhance the credibility of the AI tool.

13.  The method fails to clearly detail how discrepancies in the diagnoses from the raters were resolved. The approach must be systematic in handling disagreement over the interpretations. The process must be specified to make the diagnostics strong enough.

14.  The chosen statistical methods are not appropriately justified. Indeed, it is very relevant to explain why certain tests were chosen in light of the design of the study and the characteristics of the data to establish appropriateness for the analyses conducted.

15.  The methods section does not adequately address potential limitations related to study design, sample size, or methodological choices. A critical appraisal of the limitations is important to contextualize the findings and provide guidance for future research.

16.  While the management of data, particularly in terms of a procedure leading to the extraction of binary variables from written reports, is vaguely described, clear protocols with respect to data extraction and data management should be described in such a way that the findings would be reproducible and reliable.

17.  Although it's stated that a power simulation was performed, explicitness on methods regarding the assumptions during that process is not done. Clearly explaining the parameters using the power analysis would serve as a strong point of methodological rigor for this study.

18.  How much disagreement was there among the radiologists? Have you investigated this? How did you resolve these disagreements?

Results:

19.  While several comparisons are given along with their p-values, the exact methods of calculating these p-values are not described well. It is also relevant to establish whether any adjustments for multiple comparisons have been performed, which would directly influence the interpretation of statistical significance.

20.  Results about discrepancies between AI and radiologists could be better systematized. For example, this could be carried out by providing one summary table per diagnosis which summarizes the findings, therefore giving easier comparison and insight into the discrepancies.

21.  Results are quantitative without adequate qualitative interpretation. For example, some discussion of the clinical implications of differences between AI and radiologists would go a long way toward understanding what such findings might imply for clinical practice.

22.  Although the demographic structure of the population has been mentioned, further analysis on how this might influence the findings is still lacking. This kind of stratified analysis by age, gender, and comorbidities could give clues to the generalizability of findings.

23.  Reliability analysis suggests Cohen's Kappa and PABAK. However, the rationale for choosing these particular metrics is not provided. A discussion of why the measures are appropriate in the context of the study would strengthen the analysis.

24.  Figures and tables mentioned in the text, as can be seen in Figure 2 and Table 1, are referred to but not described in the results. Adding a small summary for the reader of what each figure/table depicts would greatly increase readability when the reader does not have immediate access to the referenced visuals.

25.  The section on inconsistencies has given examples, but it lacks an in-depth analysis of the root of such discrepancies. For example, going more in-depth into why false positives and negatives occur would enhance the discussion and yield some useful lessons.

26.  While the results here are indicative of the performance of AI in the identification of different conditions, there isn't enough discussion regarding how these findings stand in relation to existing literature. A comparison with studies that have been published previously would place results in context and give them greater meaning.

27.  The results did not show relevant gender differences in terms of diagnostic discrepancies. However, no further discussion is made on the aspect. A discussion of implications or exploring the possible reasons for the lack of such differences would give a better understanding of the results.

Discussion:

28.  This is a long discussion that is in some parts better organized using subheadings or thematic groupings of ideas to help the reader track the key points more easily.

29.  There is some repetition of results presented earlier in the paper. While a summary of key findings is important in a discussion, too much repetition detracts from the depth of analysis expected in a discussion.

30.  There is no putting the findings into the broader context of the literature. While there are some references, a more critical review of the current state of the literature on AI in radiology would help in further supporting the argument by showing more clearly why this study is needed.

31.  Although limitations are recognized, the discussion about how these limitations might impact the interpretation of results needs to go further. For example, discussing the implications of conducting a study at a single institution for generalizability could be more thorough.

32.  While the discussion raises various risks associated with over-reliance on AI, it could provide a more critical focus on what this means for clinical practice. If the findings show false positives and negatives, the discussion about the possible consequences in the patient outcomes could have been elaborated upon to make such findings relevant.

33.  While a number of reliability measures are discussed, some better explanation of why they were chosen and how they relate to the findings of the research might be useful. This will put the reported estimates of reliability in better perspective.

34.  The comparison with previous studies is somewhat incoherent. A more structured comparison synthesizing findings from similar studies would place this study within a clearer perspective in light of the existing body of research.

35.  The discussion ends with a call for further research, but it does not clearly provide recommendations or point out aspects on which further studies should focus. It would add value to the discussion if such suggestions were made.

36.  Practical implications from the results are given but can be extended further. How radiologists might incorporate AI tools into their workflow without abandoning critical oversight is a topic of discussion that could be developed into actionable insight for practitioners.

37.  The discussion essentially does not arrive at a clear conclusion that summarizes all the main findings from the study. A clearly defined conclusion would reinforce key points and provide closure to the discussion.

38.  While a lot of the discussion relates to how few AI applications have not been clinically tested, further discussion could be raised about what causes the gap and what that means for the adoption of AI in radiology.

Comments on the Quality of English Language

The manuscript has several sentences that should be simplified for easier readability, hence clarity.
Although, technical terms are necessary in scientific writing; parts of the text need to be explained so that technical jargon may be minimal. The text defines or explains very few terms in simpler words to make it more attractive to a wider audience.
There are several grammatical errors and uneasy phrasing in some places in the paper. These need a dedicated proofreading session to correct them and enhance the flow of writing in the paper.
Please check the manuscript for consistency in terms and style. It will pay off in professionalism and clarity of writing.
 Although the paper is informative, the sentence structure and length are just about the same, which makes the reading less engaging. It may bore the reader a little and he or she might lose interest.

Round 2

Reviewer 3 Report

Comments and Suggestions for Authors

Dear authors,

Thank you for revising the manuscript. Your revisions have changed the scope of the paper in terms of clarity and depth. However, I would like to give some additional points for consideration in improving the manuscript further:

Although the background of the abstract has been revised, many changes were not made. I am sure it can be recast to stay within the specified limits and yet remain fairly informative.

The discussion touches on the clinical implications; however, it would be beneficial if more cases were elaborated as to how exactly AI integration will change radiology practice concerning specific clinical practice biases, challenges, and proposed solutions.

A thorough analysis of automation bias and ways of its reduction, when generalized for clinical practice, should be performed. Training and protocol recommendations might be helpful. 
